# Selecting High-Performing and Stable Pea Genotypes in Multi-Environmental Trial (MET): Applying AMMI, GGE-Biplot, and BLUP Procedures

**DOI:** 10.3390/plants12122343

**Published:** 2023-06-16

**Authors:** Sintayehu D. Daba, Alecia M. Kiszonas, Rebecca J. McGee

**Affiliations:** 1USDA-ARS Western Wheat & Pulse Quality Laboratory, Pullman, WA 99164, USA; sintayehu.daba@usda.gov; 2USDA-ARS Grain Legume Genetics and Physiology Research Unit, Pullman, WA 99164, USA; rebecca.mcgee@usda.gov

**Keywords:** stability, WAASB, predictive assessment, postdictive assessment, cross-validation

## Abstract

A large amount of data on various traits is accumulated over the course of a breeding program and can be used to optimize various aspects of the crop improvement pipeline. We leveraged data from advanced yield trials (AYT) of three classes of peas (green, yellow, and winter peas) collected over ten years (2012–2021) to analyze and test key aspects fundamental to pea breeding. Six balanced datasets were used to test the predictive success of the BLUP and AMMI family models. Predictive assessment using cross-validation indicated that BLUP offered better predictive accuracy as compared to any AMMI family model. However, BLUP may not always identify the best genotype that performs well across environments. AMMI and GGE, two statistical tools used to exploit GE, could fill this gap and aid in understanding how genotypes perform across environments. AMMI’s yield by environmental IPCA1, WAASB by yield plot, and GGE biplot were shown to be useful in identifying genotypes for specific or broad adaptability. When compared to the most favorable environment, we observed a yield reduction of 80–87% in the most unfavorable environment. The seed yield variability across environments was caused in part by weather variability. Hotter conditions in June and July as well as low precipitation in May and June affected seed yield negatively. In conclusion, the findings of this study are useful to breeders in the variety selection process and growers in pea production.

## 1. Introduction

Pea (*Pisum sativum* L.) is used in a variety of forms including as a vegetable, when immature seeds (and sometimes pods) are consumed, as whole dry seeds, when physiologically mature seeds are consumed, and as flour and fractionated products (protein isolate and starch fraction) derived from physiologically mature seeds [1]. Breeders target peas for different market segments (yellow pea, green pea, vegetable pea, winter pea, and other market classes). A pea breeding program tailored for protein isolation is now needed due to the growing market for pea protein [2]. The USDA-ARS grain legume breeding program stationed at Pullman, Washington, focuses on developing green and yellow pea varieties, as well as winter varieties. The program began with green peas in the 1970s, was later extended to yellow peas in the 1980s, and to autumn-sown food-quality peas in 2009. High seed yield, increased protein concentration, resistance to biotic and abiotic stresses, and ease of harvesting have been the main target traits.

Breeding programs conduct multi-environment trials (METs) with advanced breeding lines to identify and recommend the best performing cultivars for a given set of environments [3]. Breeders use various statistical tools to recommend cultivars for the target environments. The breeding values of genotypes evaluated across multiple environments, estimated by best linear unbiased prediction (BLUP) from mixed models [4], can be employed in the selection process. Even if BLUP predicts genotype means more accurately, it may not show how genotypes perform across target environments. Bernardo [5] stated the two advantages of BLUP: it allows comparison of genotypes evaluated in different sets of environments and it allows use of information on relatives. The environment in which a breeding line completes its life cycle determines whether it can realize the full potential of its genotype [6]. In addition to genotypic and environmental main effects, genotype by environment interaction (GE) effect is estimated when a set of genotypes are evaluated across multiple environments. Genotypes express a range of values across environments, known as genotypes’ reaction norms [7]. When the reaction norms of at least two genotypes tend to be nonparallel or cross each other, this is referred to as GE. The presence of significant GE, especially the crossover type, has implications in breeding decisions and how genotypes respond to different environments [6,7]. 

Genotype by environment interaction can be handled in three ways, namely, (1) ignoring it, (2) reducing it, or (3) exploiting it [5]. When the GE is too small and of a non-crossover type, it can be ignored. When the GE is a crossover type and is repeatable across seasons, it can be reduced by mega-environment grouping and recommending the best cultivar for each mega-environment. Breeders may also exploit GE by identifying stable high-performing genotypes across environments. Statistical models, such as additive main effects and multiplicative interaction (AMMI) or genotype + genotype by environment interaction (GGE) biplot, have been used to address different questions relevant to plant breeding in MET, including mega-environment delineation and cultivar selection for stability and overall performance. Olivoto et al. [8] developed an index, WAASBY, that combines mean performance (denoted by Y) and stability. WAASB is the stability component of the WAASBY index and is defined as the weighted average of absolute scores from singular value decomposition of the BLUP matrix for GE generated by a linear mixed model. This procedure makes use of AMMI’s visualization capabilities and BLUP’s predictive power. 

During the breeding history of any crop, a substantial amount of data on phenotypic traits are accumulated. These data could be used to help breeders enhance their breeding program. Here, we tap into ten years (2012–2021) of yield trial data for three pea market classes (green, yellow, and winter types) from the USDA-ARS grain legume breeding program. Predictive accuracy of models can be assessed using cross-validation by dividing the data into training and validation sets [9]. Six balanced datasets were extracted to test the predictive success of BLUP and AMMI family models. Identification of potential genotypes for upcoming breeding efforts is one of the anticipated results. In this regard, the advantages of various statistical tools can be utilized. It is also crucial to compare the yields of various pea market classes and to account for how weather conditions influence pea seed yield.

## 2. Results

### 2.1. Analysis of Variance, Variance Components, and Heritability

AMMI ANOVA revealed significant (*p* < 0.01) genotypic and environmental main effects as well as genotype by environment interaction effect (Appendix A). Environment (E) contributed the largest proportion (86–93%) of the G + E + GE, while G and GE accounted for 1.9–8.2% and 5.1–8.9% of the total variation, respectively. In general, GE varies in magnitude across populations and traits [7]. Partitioning GE into signal and noise portions in the 6 datasets indicated that the signal portion was higher (52–67%) than the noise portion (33–48%). Except for the Green1820 dataset, the signal GE was more than 58% of the genotype sum of squares. These results suggest that one of the AMMI family models is suitable to analyze the datasets, except for Green1820, where AMMI analysis may not be necessary. Broad-sense heritability estimates vary according to environment and datasets (Figure 1), where it generally ranged from 0.002 to 0.870. Approximately 49% of the heritability estimates were greater than or equal to 0.5, while 26% were between 0.5 and 0.2, and the remaining 25% of heritability estimates fell below 0.2.

### 2.2. Weather and Pea Seed Yield

Best linear unbased prediction (BLUP) is useful to compare genotypes tested in different sets of environments [5]. Pea seed yield varied considerably across environments (Table 1). For green, yellow, and winter peas, mean seed yield decreased by about 85%, 87%, and 80%, respectively, in the unfavorable environment compared to the most favorable environment. In general, seed yields for all three pea market classes (green, yellow, and winter) were high in 2012, 2016, 2018, and 2020, but low in 2015 and 2021. Dayton performed poorly, while Genesee and Fairfield performed exceptionally well in most of the seasons. When only common testing environments were considered for the three market classes, the yield of the winter peas (3.5 t/ha on average) was higher than that of the spring peas (2.6 t/ha and 2.9 t/ha for green and yellow peas, respectively). 

The variability in seed yield across environments was caused in part by variations in weather conditions. Figure 2 depicts the partial least square regression (PLSR) coefficients and Pearson correlation coefficients of weather variables with environmental mean yields. Maximum temperature in June and July, especially the number of days with temperatures above 90 °C, and rainfall in May and June, especially non-rainy days in June, had a greater impact on pea productivity. Basically, hotter conditions in June and July, as well as a lack of rainfall in May and June, had a negative impact on pea seed yield.

### 2.3. Comparison of BLUP and AMMI Family Models

To evaluate the predictive success of models (BLUP and AMMI), cross-validation was applied. In all six datasets, BLUP provided better predictive success (Figure 3A–F), implying that predicted genotypic means were closer to true means when BLUP was used. When the AMMI family models were compared, the prediction accuracies were comparable. In some cases, more complex AMMI models were slightly better than less complex models. However, choosing a less complex AMMI model may be beneficial as interpretations are easier with less complex models. AMMI3 was the best model for datasets 1 (Green1218) and 2 (Green1820), and AMMI2 was the best for dataset 3 (Yellow1720). Even though complex models had slightly better predictive accuracies for 4 (Yellow1720), 5 (Winter1417), and 6 (Winter1921), AMMI3 was preferable because it was comparable to the most accurate models while being less complex.

It is also possible to select the best AMMI model based on *F*-test and cumulative explained variation, which referred to as postdictive assessment. The first 6 PCs for dataset 1 were significant at *p* < 0.05 and together accounted for 86.6% of the GE variation (Appendix A). For datasets 2, 4, 5, and 6, 4 PCs were found to be significant (*p* < 0.05); and they accounted for 82.2%, 90.6%, 92.4%, and 93.2% of the GE variations, respectively. Only 2 PCs in dataset 3 were significant (*p* < 0.05) and these 2 PCs accounted for 77.3% of the GE variation. Based on these results, AMMI6 for dataset 1; AMMI4 for datasets 2, 4, 5, and 6; and AMMI2 for dataset 3 could be selected. However, postdictive assessment may not distinguish between signal and noise portions of GE [10], which was supported by results from the current study. In all datasets, signal proportion accounted for 52–67% GE (Appendix A). Even though more principal components (two to six PCs) were found to be significant based on *F*-test, only one to three PCs accounted for most of the signal portion of GE. This result could imply that postdictive assessment captures both signal and noise parts of GE. Therefore, it might be advisable to use the signal GE as a reference to select an appropriate AMMI family model. 

In most of the cases, different AMMI family models were selected when the two model assessment methods (predictive or postdictive assessments) were applied. However, the models chosen for the datasets using predictive assessment were comparable to the models chosen with the signal GE in mind. This indicates that predictive assessment-selected models accounted for most of the signal GE. According to Gauch [10], predictive assessment offers greater accuracy when dealing with noisy data and distinguishes signal from noise as compared to postdictive assessment.

### 2.4. Selection for Overall Performance and Stability 

To identify potential lines from multi-environmental trials (METs), breeders frequently use a variety of statistical tools such as AMMI, GGE biplot, BLUP, or some combination of these. BLUP represents the breeding value of genotypes and was preferable to predict genotypes’ overall performance, as discussed in the previous section. In addition to BLUP, AMMI and GGE biplot analyses can be used to reveal yield performance and stability across environments, allowing selection for broad or specific adaptability based on breeding needs. Yield by environmental IPCA1 and yield by WAASB from AMMI, as well as “genotype ranking” GGE biplot, can be considered for this purpose. Olivoto et al. [8] developed yield by WAASB plot that presents genotypes and environments in four quadrants (Quadrants I to IV); quadrant III, for environments, and quadrant IV, for genotypes, are useful. The “genotype ranking” biplot aids in comparing test genotypes with the ideal genotype (located in the center with an arrow pointing to it), with the best genotypes being closer to the ideal genotype and the worst genotypes plotted farther away from it. We used these statistical tools to evaluate the genotypes in the six datasets.

#### 2.4.1. Dataset 1: Green1218

This dataset is comprised of 12 genotypes evaluated in 28 total environments (location–year combinations), in seasons from 2012 to 2018 (Table 2). Overall performance of genotypes presented as BLUP values indicated that G052, G031, and G002 ranked highest for seed yield (Figure 4A). Most of the genotypes exhibited crossover interactions across environments, as evidenced by the AMMI’s yield by environmental IPCA1 plot (Figure 5A). G052 and G031, for instance, performed in a crossover manner, which means that where G052 performed well, G031 performed relatively poorly, and vice versa. Across environments, G002 was consistently among the high yielders. According to WAASB by yield plot, G029, G053, and G039 were found in quadrant IV and, hence, they had above average yield with better yield stability across environments (Figure 6A). G002 was plotted in quadrant II, but closer to the cut-off line for WAASB. The other two genotypes selected based on BLUP (G052 and G031) were plotted in quadrant II, and well away from the WAASB cut-off line (Figure 6A). This implies that these two genotypes had above average yield but poor stability across environments, which agrees with the AMMI’s yield by IPCA1 results (Figure 5A). As shown in “genotype ranking” GGE biplot, G002, G053, and G029 were closer to the ideal genotype compared to the others (Figure 7A). G002 was found to be the best genotype according to BLUP, AMMI’s yield by IPCA1 plot, and “genotype ranking” GGE biplot. G053 and G029 were identified as the best genotypes with AMMI’s yield by IPCA1 plot and “genotype ranking” GGE biplot, and they were also among those genotypes with above-average mean yield according to BLUP. All the statistical tools identified G003 as a poorly performing genotype. 

#### 2.4.2. Dataset 2: Green1820

Twelve environments (location–year combinations) and nine genotypes were considered in dataset 2 (Table 2). The years were 2018 through 2020. The BLUP for genotypes revealed that G004 and G001 had the highest seed yields, while G003 had the lowest yield (Figure 4B). Referring to AMMI’s yield by IPCA1 plot, G001 remained among the high-yielding genotypes across environments (Figure 5B). In addition to G001, the top genotypes according to the WAASB by yield plot were G005 and G006 (Figure 6B). Considering the “genotype ranking” GGE biplot, G001 and G005 were relatively closer to the ideal genotype (Figure 7B), implying G001 and G005 were the best genotypes in dataset 2. All the methods identified G001 and G003 as the best and worst performing genotypes, respectively.

#### 2.4.3. Dataset 3: Yellow1216

Eight genotypes were included in dataset 3 and evaluated in fourteen environments (Table 2). G131, followed by G134, had high BLUP values (Figure 4C). G134 was among the high yielders across environments, as shown in AMMI’s yield by IPCA1 plot (Figure 5C). G128 was the only genotype that was plotted in quadrant IV for dataset 3, demonstrating that it had both high yield and stability (Figure 6C). G131 and G134, which were the best according to BLUP, were plotted in quadrant II and may be considered high yielding but with relatively low performance stability. Referring to the “genotype ranking” GGE biplot, G134 overlapped with the position of the ideal genotype (Figure 8A). G131 and G135 were also closer to the ideal genotype. In general, G134 was identified as the best genotype by three of the statistical tools (BLUP, AMMI’s yield by IPCA1 plot, and GGE biplot). G130 and G129 were identified as low yielders by all the statistical tools. 

#### 2.4.4. Dataset 4: Yellow1720

Ten yellow pea genotypes were evaluated in dataset 4 across nine environments (Table 2). BLUP determined that G129 and G130 were the genotypes with the lowest yields, as in dataset 3, while G142 and G128 were the genotypes with the highest yields (Figure 4D). G128 and G142 were among the top-five high-yielding genotypes across environments, with mean yields greater than 3.4 t/ha (Figure 5D). Three genotypes (G161, G142, and G128) were plotted in quadrant IV (Figure 6D), indicating that they performed better across the test environments. G157, G142, and G128 were the three genotypes that were closest to the ideal genotype, suggesting that these genotypes had better performance stability across environments (Figure 8B). Overall, all the statistical tools identified G128 and G142 as the best performing and G129 and G130 as poorly performing genotypes.

#### 2.4.5. Dataset 5: Winter1417

In dataset 5, 10 genotypes were evaluated across 10 environments (Table 2). G216 was the highest yielding of the tested genotypes, followed by G210 (Figure 4E). While G210 was one of the top three genotypes in all test environments, G216 displayed inconsistent performance across environments (Figure 5E). G210 and G205 were plotted in quadrant IV (Figure 6E), suggesting high yield performance across environments. G216 was plotted in quadrant II, implying that it had better overall yield performance but with low stability across environments. According to the “genotype ranking” GGE biplot, G216 followed by G205 and G210 were plotted closer to the ideal genotype as compared to the other genotypes (Figure 9A). Overall, G216 and G210 were the best genotypes in dataset 5, whereas G201 and G214 were poorly performing genotypes.

#### 2.4.6. Dataset 6: Winter1921

Seven environments and fifteen genotypes were considered in dataset 6 (Table 2). Referring to the BLUP plot, G227 was the highest-yielding genotype (Figure 4F). AMMI’s yield by IPCA1 plot identified G227 among the three high-yielding genotypes across environments (Figure 5F). As they were plotted in quadrant IV, four genotypes, G213, G227, G232, and G223, had high and consistent yields across environments (Figure 6F). The “genotype ranking” GGE biplot identified G227 as the best genotype with respect to overall performance and stability (Figure 9B). G227 was the best genotype based on all of the statistical tools used in this study, while G200 was the worst.

## 3. Discussion

Best linear unbiased prediction (BLUP) provided the best prediction of genotype means compared to any AMMI family model in all six datasets; and this result is consistent with previous research [4,8]. However, this may not be conclusive because Spoorthi et al. [11] reported that AMMI1 outperformed BLUP in prediction accuracy. In general, shrinkage property and incorporation of pedigree information maximizes the correlation between true genotypic values and predicted values [4]. There was no pedigree information included in the current study. Even though BLUP provided a better prediction of overall genotype means, it may not reveal how genotypes behave across environments. Therefore, it might be necessary to use additional statistical tools to dissect GE. The high-yielding genotypes identified by BLUP may not necessarily be winners in each environment. For example, G052 and G031 were shown to have high overall mean performance in dataset 1, but they behaved in a crossover manner across the test environments. G002 (one of the high yielders in dataset 1), on the other hand, was found to be consistent in its rank for seed yield across environments. The same could be said for the other datasets, where genotypes with high BLUP could end up being high yielders across environments or high yielders only in a specific set of environments. Combining BLUP with other statistical tools to exploit GE may thus be important in cultivar selection.

The environment (E) accounted for most of the total variation (86–93%) of pea seed yield in our study. Olivoto et al. [8] reported that E contributed highly (54%) to variation in oat yield, but GE (33%) and G (13%) also accounted for nearly half of the variation. A reaction norm is an array of values for a genotype across environments, and GE occurs when at least two of the genotypes’ reaction norms fail to be parallel or even cross one another [7]. Environmental variability, which may be manifested as weather variability, is the primary cause of genotypes’ performance differences across environments. Our research indicated that hotter temperatures in June and July, as well as a lack of sufficient precipitation in May and June, had a negative impact on pea seed yield. Tack and Holt [12] also reported that weather conditions are among the major drivers of variability of corn yield. Heat stress [13,14,15] and a lack of adequate precipitation [14] both hampered reproductive growth, resulting in flower abortion and, eventually, reduced seed yield. Temperature, precipitation, and their interactions explained approximately 60% of the variation in agricultural production in many parts of the world [16]. We also found a yield reduction of 80–87% in the worst environment versus the best environment. According to Lobell and Field [17], temperature and precipitation during the growing season influenced 30% of the year-to-year variations in productivity of the world’s six most widely produced crops (wheat, rice, maize, soybean, barley, and sorghum). The effect of weather may vary from crop-to-crop. For example, Kukal and Irmak [18] reported that increased temperature increased maize yield but decreased sorghum and soybean yields, whereas increased precipitation was useful for all the three crops. Our current findings and previous research indicate that temperature and precipitation have a significant impact on agricultural productivity.

## 4. Materials and Methods

### 4.1. Yield Trial Datasets and Measured Traits 

Initially, we used a ten-year (2012–2021) dataset from the pea advanced yield trials (AYT) that were conducted in the Palouse Region of the U.S. states of Washington and Idaho by the USDA-ARS grain legume breeding program stationed at Pullman, WA. Here, three market classes of yield trials (AYT) for spring-sown green field peas, spring-sown yellow field peas, and autumn-sown food-quality field peas were considered. A total of 127, 69, and 74 green, yellow, and winter peas, respectively, were evaluated over the course of the 10-year period (Appendix A). The complete data were used to compare the potential yields of the different pea market classes. 

To analyze the predictive success of the BLUP and AMMI family models, six balanced datasets were derived from the entire ten-year yield trial data. The six datasets comprised environments where at least eight genotypes were tested commonly (Table 2), and the list of genotypes included in each dataset were given Appendix A. In total, 2 datasets were considered for each pea market class, with 8 to 15 genotypes evaluated in 7 to 28 environments (location–year combinations). For all datasets, a randomized complete block design (RCBD) with three replications was used. Seed yield (t/ha) was considered as a response variable. We considered six datasets to see how the BLUP and AMMI family models predictively perform across different datasets. 

### 4.2. Data Analysis

Most of the data analyses were performed in R v.4.1.2 (R Core Team, 2021) using built-in functions and/or packages. For all six datasets, AMMI and GGE biplot analyses were performed using the “metan” package [19]. Broad-sense heritability estimates of seed yield were calculated as a ratio of genotypic variance to the total phenotypic variance for each pea market class within each environment [5].

For each dataset, cross-validation (CV) analyses were performed for BLUP and AMMI family models using the “cv_blup” and “cv_ammif” functions of the “metan” package, respectively [19]. The data were partitioned into a training set (two complete replications in each environment) and validation set (one complete replication in each environment). The number of resampling for cross-validation analyses was set to 1000. Models were evaluated using root-mean-square prediction difference (RMSPD), in which a smaller RMSPD signifies better prediction accuracy. Whenever models exhibited comparable prediction accuracies, the simplest model was chosen. The simplest AMMI model is defined as one with the smallest number of principal components for GE. The GE in AMMI analysis was partitioned into signal and noise portions [10,20]. The noise portion was calculated as the product of degrees of freedom for GE and mean square of residuals. Then, the signal portion of GE was calculated by subtracting the noise portion from the sum of squares for GE.

The best linear unbiased prediction (BLUP) values were generated for seed yield and plotted using the “metan” package [19] implemented in R. The WAASB was generated for each genotype using the following equation: $$\mathrm{WAASBi}=\frac{\sum_{k=1}^{p}\left|IPCA_{ik}\;\times \;EP_{k}\right|}{\sum_{k=1}^{p}EP_{k}}$$
where WAASB_i_ is the weighted average of absolute scores for *i*th genotype or environment; *IPCA_ik_* is the *i*th genotype or environment score in the *k*th interaction principal component axis (*IPCA*) score; and *EP_k_* is the variance explained by the *k*th IPCA.

A biplot was created using WAASB as the *y*-axis and seed yield (t/ha) as the *x*-axis, with genotypes and environments grouped in four quadrants [8]. The first quadrant (quadrant I) contained unstable genotypes with low productivity and highly discriminative environments, but with below average productivity. Quadrant II included unstable genotypes but with high average means and environments possessing above average productivity with high discriminative ability. Quadrant III comprised genotypes possessing low yield but with high stability. The environments in quadrant III were considered poorly productive with low discriminative ability. The genotypes in quadrant IV were highly productive with broad adaptability. Environments in quadrant IV were considered highly productive but with low discriminative ability. Quadrant II, for environments, and quadrant IV, for genotypes, were particularly useful. 

### 4.3. Analysis of Weather Data

Weather data during the crop growth period (May to August) was accessed from AgWeatherNet (https://weather.wsu.edu/). By dividing each month into three time periods (first ten days, second ten days, and all the remaining days in that month), averages (for temperature) and totals (for precipitation) were calculated. For the months of June and July, we also counted the number of days without rain and the number of days with temperature ≥ 90 °C. Weather variables (temperature and precipitation) during the crop growth period (May to August) were used as explanatory variables in partial least squares regression. The acronyms for the explanatory variables are made up of the weather variables (MaxT and RF), the months (May, June, and July), and the numerals 1, 2, 3, 0, and 90. MaxT and RF stand for maximum temperature and rainfall, respectively. The days 1 through 10, 10 through 20, and days 21 and beyond are denoted by the digits 1, 2, and 3, accordingly. RF0 represents the number without rain and MaxT > 90 denotes the number of days with a maximum temperature more than 90 °C in the months of June and July. Pearson correlation coefficient analysis was performed for seed yield with weather variables. Weather drivers of seed yield variability were studied using the regression and correlation coefficients.

## Figures and Tables

**Figure 1 plants-12-02343-f001:**
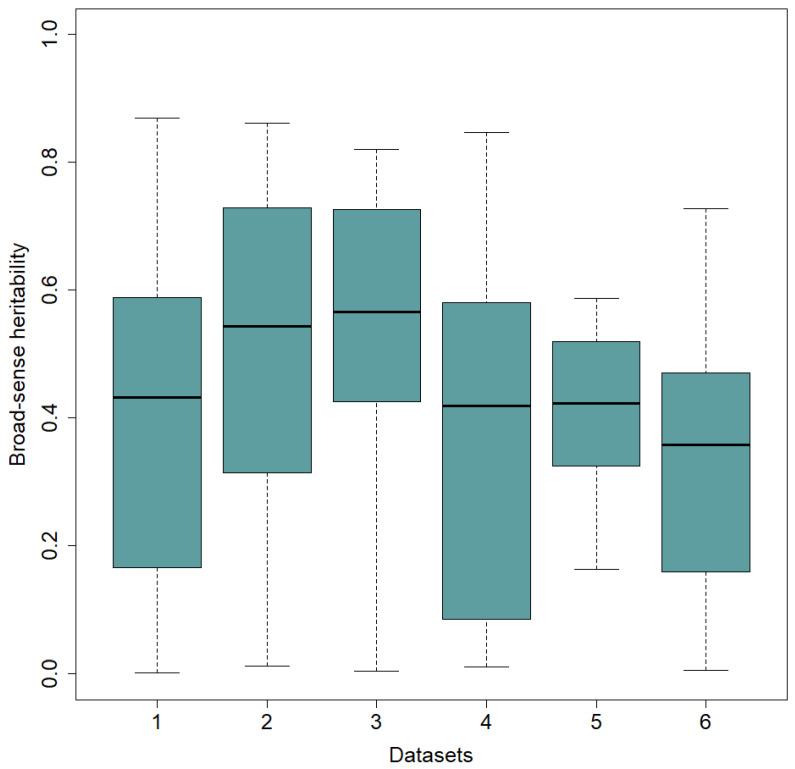
Broad-sense heritability estimates in the six datasets: dataset 1, 2, 3, 4, 5, and 6 represent Green1218, Green1820, Yellow1216, Yellow1720, Winter1417, and Winter1921, respectively.

**Figure 2 plants-12-02343-f002:**
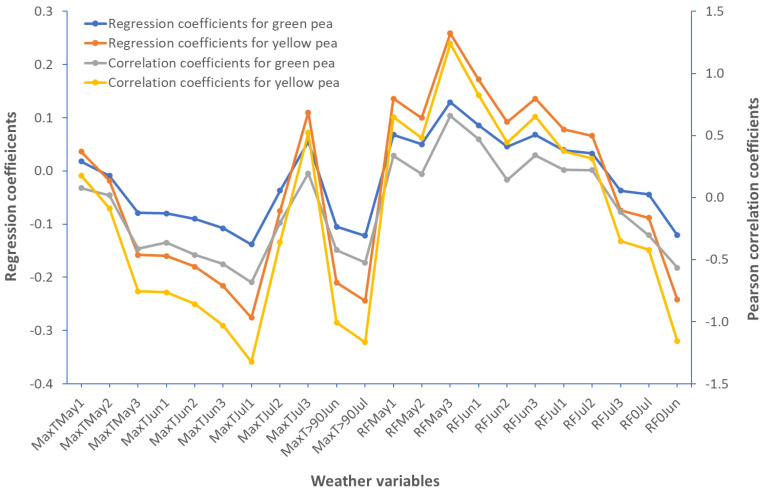
Regression coefficients from partial least square regression (PLSR) and Pearson correlation coefficient analyses. MaxT = maximum temperature, MaxT > 90 = number of days with maximum temperature greater than 90 °C, RF = rainfall, RF0 = number of days without rainfall in that month; the number associated with months (e.g., May1) indicates the parts of the month (example: May1 = the first 10 days of May, May2 = days from May 11 to May 20, May3 = days after May 20).

**Figure 3 plants-12-02343-f003:**
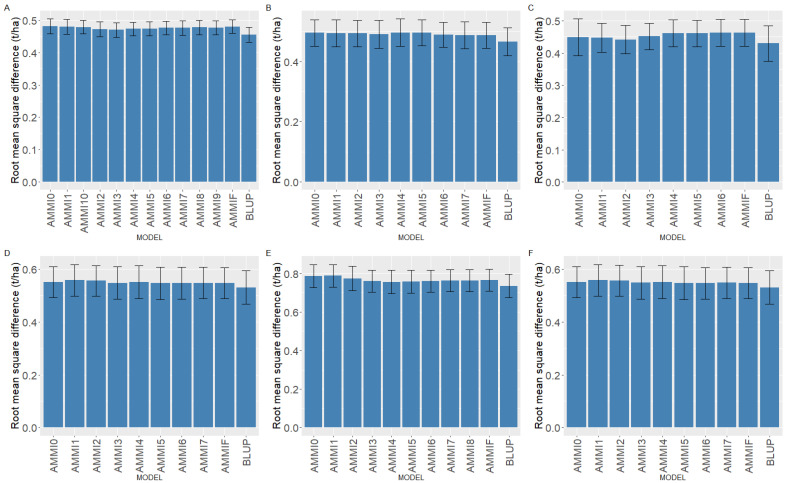
Predictive assessment using cross-validation of best linear unbiased prediction (BLUP) and AMMI family models for (**A**) dataset 1, (**B**) dataset 2, (**C**) dataset 3, (**D**) dataset 4, (**E**) dataset 5, and (**F**) dataset 6. AMMI0 to AMMF represent AMMI family models, with AMMI0 incorporating no principal components and AMMIF incorporating all the possible principal components.

**Figure 4 plants-12-02343-f004:**
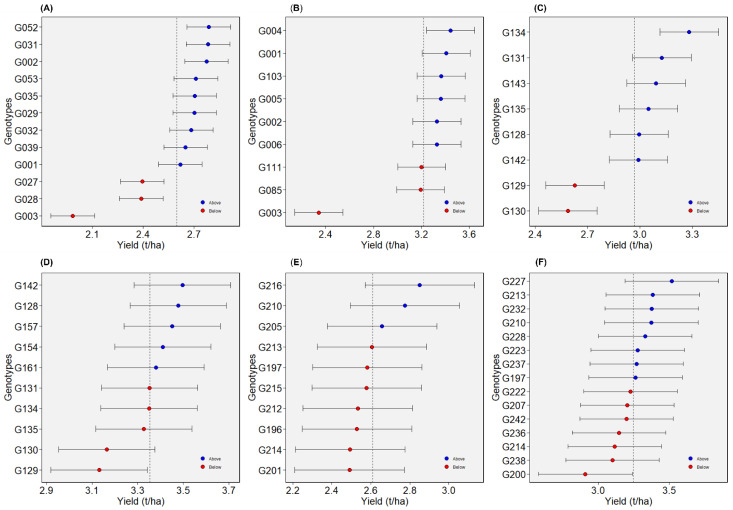
Best linear unbiased prediction (BLUP) for evaluated genotypes for (**A**) dataset 1, (**B**) dataset 2, (**C**) dataset 3, (**D**) dataset 4, (**E**) dataset 5, and (**F**) dataset 6.

**Figure 5 plants-12-02343-f005:**
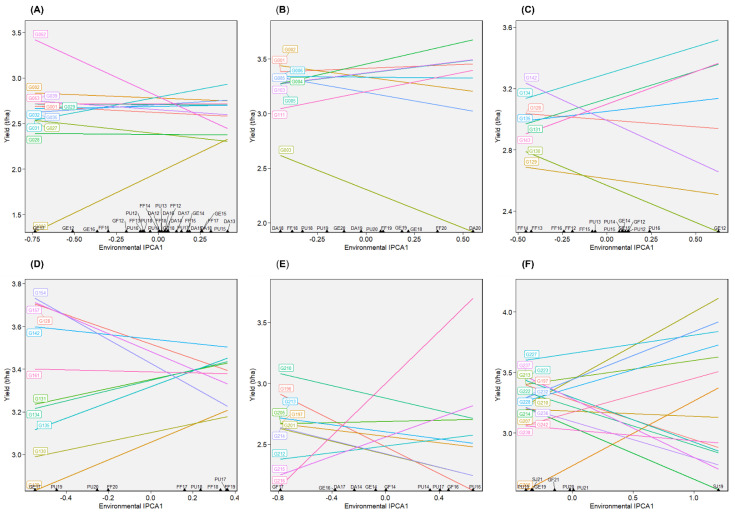
Performance of genotypes across the environments in the six datasets presented in AMMI’s yield by environmental IPCA1 plot for (**A**) dataset 1, (**B**) dataset 2, (**C**) dataset 3, (**D**) dataset 4, (**E**) dataset 5, and (**F**) dataset 6.

**Figure 6 plants-12-02343-f006:**
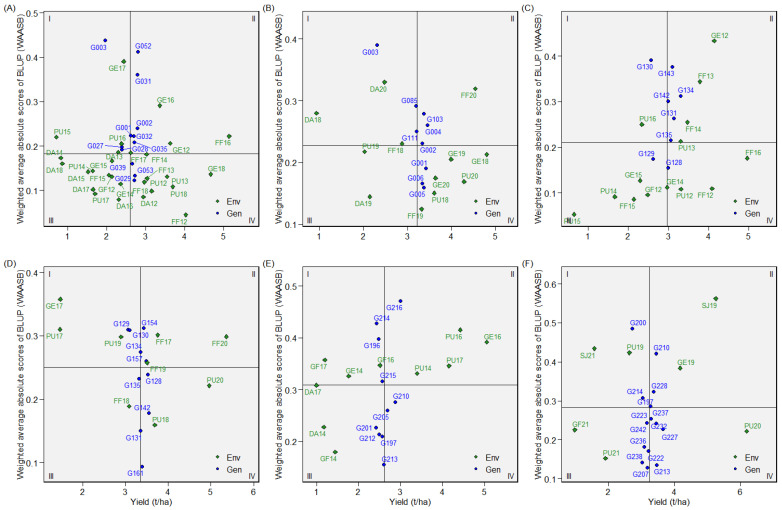
WAASB by yield plot presenting genotypes’ overall performance and stability as well as environments’ productivity and discriminative ability for (**A**) dataset 1, (**B**) dataset 2, (**C**) dataset 3, (**D**) dataset 4, (**E**) dataset 5, and (**F**) dataset 6.

**Figure 7 plants-12-02343-f007:**
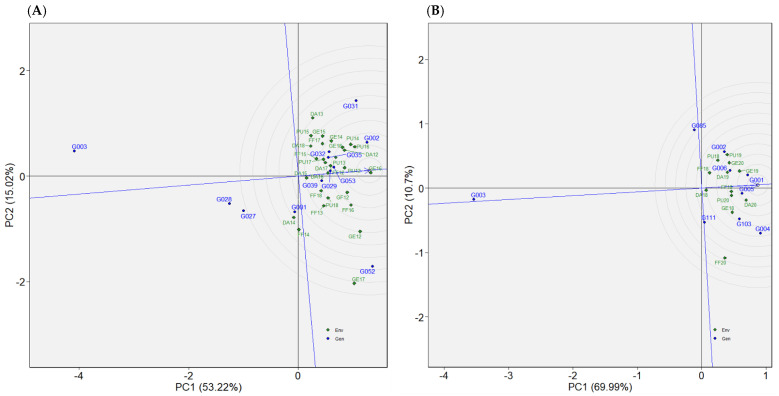
Genotype ranking GGE biplot presenting comparison of evaluated genotypes relative to the ideal genotype represented by the center of concentric circles and an arrow pointing to it. The graphs represent (**A**) dataset 1 and (**B**) dataset 2.

**Figure 8 plants-12-02343-f008:**
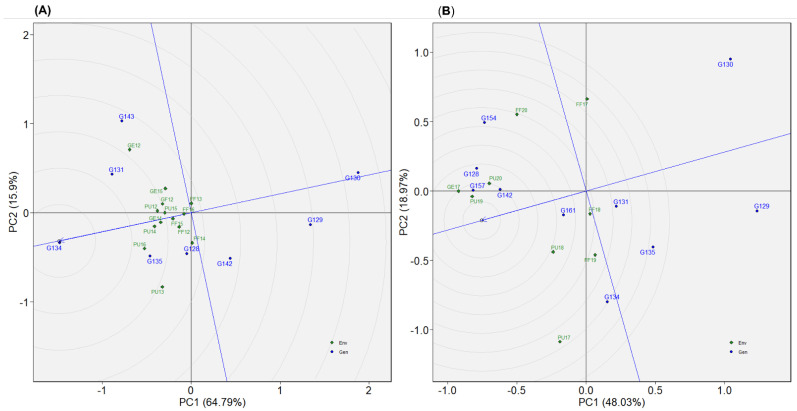
Genotype ranking GGE biplot presenting comparison of evaluated genotypes relative to the ideal genotype represented by the center of concentric circles and an arrow pointing to it. The graphs represent (**A**) dataset 3 and (**B**) dataset 4.

**Figure 9 plants-12-02343-f009:**
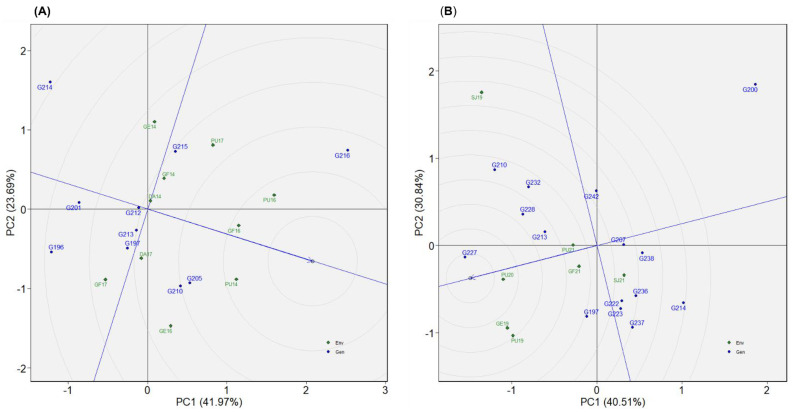
Genotype ranking GGE biplot presenting comparison of evaluated genotypes relative to the ideal genotype represented by the center of concentric circles and an arrow pointing to it. The graphs represent (**A**) dataset 5 and (**B**) dataset 6.

**Table 1 plants-12-02343-t001:** Mean and coefficient of variation (CV) of seed yield (t/ha) for genotypes tested in each environment (location–year interaction) for the three pea market classes.

Environments	Green Peas		Yellow Peas		Winter	
	Mean	CV	Mean	CV	Mean	CV
DA2012	2.9	9.2			2.8	19.0
DA2013	2.2	9.5				
DA2014	0.9	11.2			1.2	19.2
DA2015	1.5	9.5				
DA2016	2.3	9.2				
DA2017	1.5	10.2			1.1	19.8
DA2018	0.9	11.4			2.1	19.4
DA2019	2.0	9.8			1.3	22.4
DA2020	2.3	10.7				
FF2012	4.0	9.0	4.0	8.6		
FF2013	2.9	9.4	3.7	9.5		
FF2014	2.9	8.6	3.3	9.4		
FF2015	2.2	8.5	2.1	10.0		
FF2016	4.9	9.6	4.9	8.2		
FF2017	2.6	9.1	3.7	9.7		
FF2018	2.8	8.8	3.1	8.7		
FF2019	3.2	9.8	3.4	8.7		
FF2020	4.4	9.0	5.3	9.4		
FF2021	1.1	10.1	2.1	10.0		
GE2012	3.7	9.6	4.1	7.8		
GE2014	2.2	9.0	3.0	8.2	1.5	20.5
GE2015	2.2	8.5	2.3	9.9		
GE2016	3.3	10.2			4.7	18.6
GE2017	2.7	14.7	1.6	20.4		
GE2018	4.6	8.9				
GE2019	3.9	9.4			4.0	18.3
GE2020	3.5	9.6				
GF2012	2.2	9.8	2.5	8.9	4.6	19.2
PU2012	3.2	9.3	3.3	8.5	3.7	18.6
PU2013	3.4	9.1	3.4	7.8	1.7	20.4
PU2014	1.8	10.5	1.7	10.8	3.1	20.2
PU2015	0.7	11.3	0.7	20.4		
PU2016	2.4	9.3	2.4	8.2	3.9	19.5
PU2017	1.7	9.1	1.5	10.1	4.0	18.9
PU2018	3.5	8.9	3.6	9.0	4.0	20.2
PU2019	2.1	9.7	2.8	10.4	2.4	20.0
PU2020	4.0	8.8	4.8	8.6	5.6	18.8
PU2021	1.8	9.5	1.5	12.6		

**Table 2 plants-12-02343-t002:** Datasets of yield trials for the three types of pea (green, yellow, and winter) with the genotypes and test environments.

Type	Environments	Number of Genotypes
Dataset 1: Green1218: Green peas	DA, FF, and PU (2012–2018), GE (2012, 2014–2018), GF (2012)	12
Dataset 2: Green1820: Green peas	DA, FF, GE, and PU (2018–2020)	9
Dataset 3: Yellow1216: Yellow peas	FF and PU (2012–2016), GE (2012, 2014, 2015), GF (2012)	8
Dataset 4: Yellow1720: Yellow peas	FF and PU (2017–2020) GE (2017)	10
Dataset 5: Winter1417: Winter peas	GF and PU (2014, 2016, 2017), DA (2014, 2017), GE (2014, 2016)	10
Dataset 6: Winter1921: Winter peas	PU (2019–2021), GE and SJ (2019, 2021)	15

DA = Dayton, FF = Fairfield, GE = Genesee, GF = Garfield, PU = Pullman, and SJ = St. John.

## Data Availability

Summarized data were provided in supplemental Appendix A.

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
