# Peer review of "Selecting High-Performing and Stable Pea Genotypes in Multi-Environmental Trial (MET): Applying AMMI, GGE-Biplot, and BLUP Procedures"

_plants, 2023, doi:10.3390/plants12122343_

Round 1
Reviewer 1 Report
I have gone through the paper entitled "Selecting High Performing and Stable Pea Genotypes in Multi-Environmental Trial (MET): Applying AMMI, GGE-biplot, and BLUP Procedures". The authors used data from advanced yield trials of three classes of peas (green, yellow, and winter peas) to analyze and test key aspects fundamental to pea breeding. Six balanced datasets were used to test the predictive success of BLUP and AMMI family models. Such studies are of great significance for breeders in the variety selection process and growers in pea production. The manuscript is well-structured and presents a lot of data. This is indeed an interesting study in line with the scope of the journal. I have no qualms about recommending this manuscript for acceptance. However, some changes need to be addressed before any further steps can be taken.
(1). Most of the keywords are already mentioned in the title, please replace them.
(2). Line 36= Please mention some names of winter varieties.
(3). Line 192 = Figures should be self explanatory. Please define MaxT, RF, RF0 in the caption.
(4). Line 208= I suggest improving the quality of figures.
Reviewer 2 Report
Pea is an important nutritious plant with a variety of forms. In this study, with a large amount of data of ten years, predictive models were assessed and potential genotypes were identified. There are some suggestions as following:
1. Table 1 is a little messy, it is recommended to furthur align and organize, and to list the number of genotypes in each data set.
2. In materials and methods, it is recommended to list and describe the analysing methods of environmental factors on seed yield separately. The description order of the materials and methods in the article is inconsistent with that in the results section, and the hierarchy is confusing.
3. Why not consider weather data as a fixed effect and genotype as a random effect for multi-environment prediction analysis?
4. Is the portion of the phenotypic data whose heritability is lower than 0.2 considered to be deleted, and how to cause the lower heritability data?
5. The format of Table 2 is incorrect and needs to be rearranged. If it needs to be described separately. It is recommended to make a three-line table. And the abbreviations in the table can be added.
6. Figure 3 is too big, the abscissa scales of the six boxplots should be consistent, and the legend does not explain A-F.
7. The format of the references is inconsistent, please modify.
8. Why is it divided into 6 data sets, which is not explained in the article.
9. In this paper BLUP provides a better prediction of the overall genotype mean, but it may not reveal how the genotype behaves in different environments. Why it is believed that AMMI and GGE can reveal, whether these two can improve the prediction of cross-environment is also not tested in this paper? At the same time, GE effects can also be added to BLUP for analysis.
